# Prodrug Therapies for Infectious and Neurodegenerative Diseases

**DOI:** 10.3390/pharmaceutics14030518

**Published:** 2022-02-26

**Authors:** Milica Markovic, Suyash Deodhar, Jatin Machhi, Pravin Yeapuri, Maamoon Saleh, Benson J. Edagwa, Rodney Lee Mosley, Howard E. Gendelman

**Affiliations:** 1Department of Pharmacology and Experimental Neuroscience, College of Medicine, University of Nebraska Medical Center, Omaha, NE 68198-5880, USA; mmarkovic@unmc.edu (M.M.); suyashsanjay.deodhar@unmc.edu (S.D.); jatin.machhi@unmc.edu (J.M.); msaleh@unmc.edu (M.S.); benson.edagwa@unmc.edu (B.J.E.); rlmosley@unmc.edu (R.L.M.); 2Department of Pharmaceutical Sciences, College of Pharmacy, University of Nebraska Medical Center, Omaha, NE 68198-5880, USA; pravin.yeapuri@unmc.edu

**Keywords:** prodrugs, drug derivatization, neurodegenerative disorders, infectious diseases, human immunodeficiency virus (HIV), SARS-CoV-2, hepatitis, HIV-associated neurocognitive disorders (HAND)

## Abstract

Prodrugs are bioreversible drug derivatives which are metabolized into a pharmacologically active drug following chemical or enzymatic modification. This approach is designed to overcome several obstacles that are faced by the parent drug in physiological conditions that include rapid drug metabolism, poor solubility, permeability, and suboptimal pharmacokinetic and pharmacodynamic profiles. These suboptimal physicochemical features can lead to rapid drug elimination, systemic toxicities, and limited drug-targeting to disease-affected tissue. Improving upon these properties can be accomplished by a prodrug design that includes the careful choosing of the promoiety, the linker, the prodrug synthesis, and targeting decorations. We now provide an overview of recent developments and applications of prodrugs for treating neurodegenerative, inflammatory, and infectious diseases. Disease interplay reflects that microbial infections and consequent inflammation affects neurodegenerative diseases and vice versa, independent of aging. Given the high prevalence, personal, social, and economic burden of both infectious and neurodegenerative disorders, therapeutic improvements are immediately needed. Prodrugs are an important, and might be said a critical tool, in providing an avenue for effective drug therapy.

## 1. Introduction

Prodrugs are inactive drug derivatives that undergo chemical and or enzymatic-mediated hydrolysis into an active parent drug in plasma and tissues [1]. The biotransformation is designed to overcome suboptimal physicochemical drug features, rapid drug metabolism, poor absorption and distribution, and improve the overall pharmacokinetic and pharmacodynamic (PK and PD, respectively) profiles. Such improvements could also limit the systemic toxicities through targeted drug delivery [1,2,3,4,5]. Developing new drugs is time limited and costly [6]. Prodrugs can speed up drug development by improving the drug’s physicochemical characteristics while reducing the developmental cost. Indeed, this approach is broadly used in all phases of the drug development from newly developed drug entities to commercialization [7]. Indeed, up to 10% of commercially available medicines are prodrugs [8]. Important considerations in prodrug design and development include several of the following:

**Inherent native drug features**. The physiochemical properties and nature of the functional groups that are present on the native drug will determine the drug conjugation sites, bond strength, and any needed molecular modifications.

**Auxiliary grafts**. The promoieties for drug conjugation are selected for grafting based on hydrophobicity, pH, and protein binding [9]. The graft must be biocompatible with the microenvironment and excreted upon prodrug cleavage [3,10].

**Linker**. The linker, or the connecting segment, is typically located between the native drug and the promoiety and is designed to affect the liberation of the native drug from the prodrug assembly. It can also impact the overall stability of the prodrug and the formulation process along with the compound conformation that affects drug-target site interactions [11,12,13,14,15].

**Formulation**. This serves to optimize the prodrug fate, linker cleavage, native drug release, bioavailability, and biodistribution. The prodrug may be encased [16], conjugated [17], or even incorporated into the delivery devices including transdermal patches, microneedles [18], or implants [19]. Thus, the prodrug development platform provides a versatile system for a wide range of therapeutics and for a spectrum of administration routes that include enteral and parenteral routes. This approach remains an important tool for modifying already existing, as well as newly developed drugs, and hereinafter we will discuss in detail the use of this approach for some of the most challenging global health conditions.

Infectious and neurodegenerative diseases are the leading causes of disease morbidity and mortality worldwide [20]. Relationships between infectious and neurodegenerative diseases that include Alzheimer’s and Parkinson’s diseases (AD and PD) and amyotrophic lateral sclerosis (ALS) can be age-independent [21]. According to the World Health Organization (WHO), in 2019, AD and other forms of dementia ranked as the seventh leading cause of death in the world. Indeed, AD alone accounts for 60–70% of all diagnosed dementia cases, and the prevalence of AD cases is expected to exceed 130 million by 2050 [22]. Women are affected by AD disproportionately (65% of AD deaths are females) [23,24]. PD is only second to AD as the most common neurodegenerative disorder with a global prevalence of over six million [25]. Infectious diseases, likewise, are amongst the most common diseases worldwide and disproportionately affects low-income countries with high death rates. In 2019, lower respiratory tract infections and diarrheal diseases were ranked in the top ten causes of death worldwide by the WHO. Moreover, globally, there are 300–500 million cases of malaria and 333 million of sexually transmitted diseases that included syphilis, gonorrhea, chlamydia, and trichomonas. Another 40 million cases were attributed to infections with the human immunodeficiency virus type-one (HIV-1), cholera, and tuberculosis [26]. This is in addition to the 340 million cases of COVID-19, referenced by World Health Organization (WHO) Coronavirus (COVID-19) Dashboard [27]. Based on disease prevalence, a limited numbers of needed drugs have been US Food and Drug Administration (FDA) approved [8]. High-throughput drug screening could overcome this limitation. Nevertheless, this method of drug discovery leads to drug candidates with undesirable physicochemical features (poor permeability, poor solubility, and poor site targeting). Together with the fact that antimicrobial drugs face obstacles when it comes to drug resistance, and several drugs that were developed to combat neurodegenerative diseases face additional obstacles that are linked to blood brain barrier penetrance and site specificity, improvements in the drug development process are desperately needed. This review serves to highlight one part of drug development which is the development of prodrugs. We posit that this pathway can aid in drug development, serving to improve disease treatment and prevention.

## 2. Prodrug Strategies for Neurodegenerative Diseases

### 2.1. Current and Prospective Therapies for Neurodegenerative Diseases

Neurodegenerative diseases include but are not limited to, AD, PD, ALS, Huntington’s disease (HD), and infections of the nervous systems in which neurons and glial cells are injured then lost [28]. The symptoms are reflective of the disease and the area(s) of the central nervous system (CNS) that are injured, which include motor, sensory, speech, and/or cognitive impairments amongst others [29]. An underlying pathology in neurodegenerative diseases include cell-associated aggregation of misfolded and biochemically-modified proteins. These include, for example, amyloid-β and tau, α-synuclein, and superoxide dismutase 1 and TAR DNA-binding protein 43 (TDP-43) in AD, PD, and ALS, respectively [30,31]. Moreover, each of the disorders is lifelong and currently without cure. Current treatment options primarily treat the disease symptoms, but fail to slow the progression or address the causation. Novel neuroprotective, disease-modifying, replacement, and immune transformative therapies are being developed to slow neurodegenerative processes, but have not yet reached broad patient use [30,32], despite vigorous attempts to affect the disease outcomes. One translational avenue to improve brain disease outcomes is to affect the drug-targeting, pharmacokinetics, pharmacodynamics, and physicochemical properties that can be achieved through the generation of prodrugs [33]. This improvement may extend the drug effect, evade drug resistance, or prolong the therapeutic window narrowing until new disease-transformative therapies become available. The most compelling cause of multiple sclerosis is Epstein-Barr virus although absolute associations and specific antiviral therapies remain in cause and development. 

AD is the most common neurodegenerative disorder, and is the most common cause of dementia. It is closely associated with aging as the elderly population is most disproportionately afflicted. The disorder is mainly characterized by the deposition of misfolded self-protein aggregates, extracellular amyloid beta (Aβ) plaques, and intracellular neurofibrillary tangles [34,35]. Prior to the FDA approval of the Aβ-targeting antibody, aducanumab, in 2021, few therapeutics options were available. In larger measures, this therapeutic received controversial accelerated approval for conditions, but with a recommendation to re-examination [36,37,38].

PD is the second most common neurodegenerative disorder and the most common neurodegenerative movement disorder [39]. Motor symptoms reflect the progressive degeneration of dopaminergic neurons within the substantia nigra (SN) pars compacta with projections to the caudate nucleus [40]. This leads to diminished dopamine levels, which manifest as resting tremor, bradykinesia, rigidity, and gait dysfunction. Numerous, nonmotor symptoms are common and include those that are associated with pain, fatigue, low blood pressure, bladder and bowel dysfunction, sleep disorders, and dysphagia. Neuropsychiatric symptoms that include hallucinations and delusions are also common [40]. Despite efforts in understanding the pathobiology of PD, the available disease-modifying drugs focus principally on restoring the brain dopamine levels [41].

For both AD and PD, current therapeutic research has been focused on developing better means to combat disease [42]. Indeed, with no curative treatments available, intense interests exist towards improving therapies that already exist. One strategy of interest is the development of prodrugs. In recent years intense efforts have been made including antibody targeting; nanomedicine-linked drug delivery that is associated with the use of carbon nanotubes and lipid nanoparticles. These agents can overcome drug membrane permeability and penetration as well as improving drug delivery, solubility, and stability, while reducing toxicity. However, an equally encouraging vehicle for drug delivery is through the development of prodrugs. Creating prodrugs serves to enhance the physical and chemical parent drug properties, while overcoming the delivery barriers for an active drug. Prodrugs can serve the field of neurodegenerative pharmacology as they contain existing therapeutic parent drugs for AD and PD, which can be improved through the prodrug approach. This approach will lower the side effects and hydrophilicity, while improving blood-brain-barrier (BBB) permeability with higher brain–blood drug concentrations and improved efficacy. Each of these parameters are essential for better patient disease outcomes [43]. Hereinafter, we present an overview of prodrugs that are used for the two most common neurodegenerative diseases, AD and PD.

### 2.2. Prodrugs for Alzheimer’s Disease

Memantine is a second-line therapeutic agent that is used for the management of AD. It is an *N*-methyl-*D*-aspartate (NMDA) receptor antagonist and blocks excessive and prolonged glutamate exposure; more specifically, it blocks calcium channel flow by extrasynaptic-, but not synaptic-NMDA receptors. This atypical behavior allows memantine to attenuate excitotoxicity while preserving glutamatergic synaptic functions to offer clinical tolerability [44,45]. A sulfide analogue of memantine was developed as the prodrug, memit, by replacing the free amino group of memantine with hydrogen sulfide (H_2_S) (Table 1). H_2_S is produced endogenously from the amino acids L-cysteine and homocysteine (HCy). This is aided by several enzymes (cystathionine β-synthase, cystathionine γ-lyase, and 3-mercaptopyruvate sulfurtransferase together with cysteine aminotransferase). Moreover, it interacts with the NMDA receptor directly, through the sulfhydration of cysteine residues, and indirectly, by regulating the intracellular Ca^2+^. Memit readily converts into the memantine by releasing H_2_S through a cysteine-dependent mechanism. This novel hybrid molecule demonstrated protective effects against neuronal inflammation as well as diminished reactive oxygen species (ROS) production. Memit also protected human neuronal cells and rat microglia against Aβ oligomer-induced toxicity [46]. Further in vivo studies will confirm the synergic effects of a hydrogen sulfide-releasing moiety in combination with other native drugs for the treatment of AD.

Another example of the prodrug that was developed for improving existing AD drugs is galantamine benzoate (Gln-1062, Memogain^®^, Galantos Pharma GmbH, Mainz, Germany), an inactive lipophilic prodrug of galantamine. It liberates the parent drug on cleavage by carboxyesterases in the brain. It is a specific, competitive, and reversible acetylcholinesterase inhibitor which shows mild cognitive benefits in AD patients. In ferrets, Gln-1062 avoided peripheral side effects and offered more than 15-fold higher bioavailability in the brain compared to native galantamine at a human equivalent dose. In a scopolamine-induced amnesic mouse model, Gln-1062 completely reversed memory impairments at a three-times lower dose compared to the parent drug [47]. In the 5xFAD mouse model, Gln-1062 reduced the plaque load and improved memory functions. In a first-in-human study, Gln-1062 was found to be safe and well tolerated up to 22 mg, twice daily dose upon intranasal administration compared to oral galantamine. Due to higher BBB penetration and rapid conversion into the parent drug, Gln-1062 was responsible for improved cognitive functions in AD patients as measured by NeuroCart, a battery of neuropsychological and neurophysiological tests [48].

Apart from the classical etiological factors, the role of growth factors is frequently studied in AD pathogenesis, given their important role in regulating neuronal vitality and differentiation. Brain-derived neurotrophic factor (BDNF) and other neurotrophic growth factors exert pleotropic effects by activating cognate tropomyosin receptor kinase B (TrkB) receptors [49,50]. Indeed, BDNF exerted protective effects against AD pathogenesis in animal models by reducing amyloid load and improving cognitive functions [51]. Despite encouraging preclinical outcomes, several clinical trials with recombinant BDNF failed, presumably due to low bioavailability and poor delivery inside the brain. A potent TrkB receptor agonist, 7,8-dihydroxyflavone (7,8-DHF) that mimics BDNF functions was identified as a potential agent that exhibits selective and high affinity for TrkB receptor [52]. Although, systemic administration of 7,8-DHF activated the TrkB receptor in the brain and induced BDNF-like behavioral improvements in a variety of animal models, it exhibits low bioavailability which hinders clinical translatability [52,53]. R13 is a prodrug derivative of 7,8-DHF and was derivatized by masking the hydroxyl group by carbamate functional group on the catecholamine ring in 7,8-DHF. These modifications prolonged the half-life and increased the oral bioavailability more than ten-fold compared to the parent drug (Table 1). The highly lipophilic nature of R13 significantly increased brain exposure and attenuated amyloid deposition and memory impairments in 5xFAD mice in a dose-dependent manner [54].

The role of neuroinflammation in AD pathogenesis has been well-established over the last decades [55]. Aβ aggregates inside the brain activate microglia leading to the secretion of proinflammatory cytokine milieu that ultimately affect neuronal viability and cognitive functions [54]. *N,N′*-diacetyl-*p*-phenylenediamine (DAPPD) is a potent anti-inflammatory molecule that is reported to alleviate microglial activation and restore phagocytic capabilities through modulating the NF-κB pathway and NLRP3 inflammasome expression in AD mice brains, leading to reduced amyloid deposition and memory improvement [56,57]. However, DAPPD exhibits limited biological applicability due to low solubility in aqueous medium and poor BBB permeability. The glucose transporter system (GLUT1) transports glucose across the BBB. A Glu-DAPPD prodrug was designed by linking the glucose moiety to DAPPD to facilitate brain entry. Glu-DAPPD lowered amyloid aggregation and improved memory functions in the APP/PS1 double transgenic mouse model of AD [58].

Tramiprosate (homotaurine, 3-amino-1-propanesulfonic acid, or 3-APS) is a small molecule that inhibits oligomerization and aggregation of Aβ [59]. Tramiprosate was shown to be neuroprotective against Aβ toxicity through the activation of β-aminobutyric acid-A (GABA-A) receptors [60]. Tramiprosate also reduced amyloid burden in hAPP-TgCRND8 transgenic mice [61]. In clinical studies, orally administered tramiprosate reduced Aβ_42_ levels in the cerebrospinal fluid (CSF) of AD patients [62]. However, in a Phase 3 clinical trial, tramiprosate failed to show benefit in AD patients [60]. It was identified that tramiprosate exhibits two major deficiencies: high variability in patients’ bioavailability, most likely due to gastrointestinal metabolism, and incidences of nausea and vomiting. To overcome such limitations, a novel valine-conjugated prodrug of tramiprosate, (13-(*L*-valyl) amino-1-propanesulfonic acid, ALZ-801), was designed (Table 1). Orally administered ALZ-801 is absorbed from the gastrointestinal tract and releases tramiprosate by cleavage by hepatic or plasma amidases. This offeres significantly improved bioavailability over the parent drug. Doses of 265 mg twice daily of ALZ-801 achieved a plasma concentration that was equivalent to 150 mg of tramiprosate and improved cognition and function in apolipoprotein E4/4 homozygous AD patients in a Phase 1 clinical trial [63]. Currently, AD patients are being recruited for a Phase 3 clinical trial with ALZ-801.

Successful prodrug design was shown to alleviate neuroinflammation, improve efficacy, and accomplish enhanced safety and delivery of existing therapeutic agents to combat this devastating disease. This is particularly important in conditions, such as AD, with limited therapeutic alternatives.

### 2.3. Prodrugs for Parkinson’s Disease (PD)

Levodopa (l-3,4-dihydroxyphenylalanine, *L*-DOPA) (LD), is a prodrug of dopamine and remains the gold standard for the treatment of PD [64], despite numerous drawbacks. Some of LD shortcomings are that it provides only symptomatic therapy, it has a short half-life due to decarboxylation by DOPA decarboxylase, and patients become retractile following years of treatment. LD also causes frequent side effects, such as nausea, vomiting, and orthostatic hypotension [65]. To reduce metabolic instability, peripheral side effects, and improve the blood-brain bioavailability, LD is usually administered together with peripheral decarboxylase inhibitors such as benserazide and carbidopa.

Over the years, several LD prodrugs were designed to overcome problems with LD bioavailability and peripheral metabolism [65,66,67]. Optimal LD prodrugs should replicate the physiological striatal dopamine (DA) levels, produce long-term safety/tolerability, delay narrowing of the therapeutic window, and at the same time, delay disease progression; however, such a prodrug has yet to be identified. Extensive research is underway and numerous examples of prodrug development for PD have been reported in the literature. For instance, actively transported LD prodrug, XP21279, was studied in PD patients who experience motor fluctuations. The sustained-release of the LD prodrug was actively absorbed by high-capacity intestinal transporters and rapidly converted to levodopa via enzyme carboxylesterases (Figure 1). Thus, XP21279-carbidopa sustained-release bilayer tablets were developed to overcome high rates of prodrug conversion leading to pharmacokinetic limitations by providing greater continuous exposure [68]; nevertheless, further prodrug development was halted due to performance issues in attaining significant primary outcomes in clinical trials.

DA is rapidly metabolized following oral administration, thus cannot enter the BBB via passive diffusion and necessitates the development of a prodrug approach to overcome this obstacle. Amino acid prodrugs of dopamine carry the cationic drug into the brain by the *L*-type amino acid transporter 1 (LAT1, SLC7A5); a promising target for brain drug delivery of poorly penetrating drugs using a phenylalanine promoiety that is attached to the DA parent drug through an amide bond [69].

Indeed, as mentioned earlier, the chemical structures of prodrugs play an important role in DA prodrug development. In general, studies show that amide LD prodrugs are more stable than ester-containing prodrugs. In fact, amide, cyclic, and peptidyl LD prodrugs increased the enzymatic/hydrolytic stability and consequently improved the PK parameters [66], while monomeric- and dimeric-amide prodrugs enhanced the blood brain barrier (BBB) penetration and better central nervous system (CNS) pharmacokinetics. In addition, attaching LD to sugars has been used to exploit glucose transport mechanisms into the brain [70]. Examples of LD prodrugs have been designed to improve physicochemical characteristics that enable suitable formulation and delivery, which in turn, may accomplish higher LD selectivity/targeting towards the striatum. For instance, *L*-DOPA methyl ester hydrochloride (LDME), an ester prodrug of LD, was designed for intranasal delivery, a route that ensures direct brain penetration, since it bypasses the limitations of the BBB. Administration of the nasal powder containing LDME reaches up to 80% bioavailability [71]. Another example of successful prodrug formulation is ABBV-951 (foslevodopa/foscarbidopa), which is a formulation of levodopa/carbidopa prodrug in development for the treatment of motor complications for patients with advanced PD (aPD) [72,73]. The solubility of ABBV-951 allows for continuous subcutaneous (s.c.) infusion (as opposed to usual oral), which is able to provide a stable levodopa exposure over 72 h [74] and is currently being evaluated in a Phase 3 study (NCT04380142) in patients with aPD [75]. Another interesting example of levodopa/carbidopa treatment is the device-aided administration to PD patients via intestinal gel [76].

Besides DA and LD derivatives, a novel prodrug derivative of geraniol and ursodeoxycholic acid (UDCA) is a PD drug candidate [77]. Geraniol and UDCA ester conjugate produces a GER-UDCA prodrug which undergoes esterase hydrolysis. Geraniol has a strong anti-inflammatory effect, which promotes the survival of dopaminergic neurons through increased levels of antioxidant enzymes and neurotrophic factors as well as reduced levels of apoptotic factors [78]. This was combined with the mitochondrial rescue effect that is associated with UDCA, which countered interactions of the glucocorticoid receptor and increased phosphorylation of the Akt protein that leads to mitochondrial-dependent programmed cell-death [79]. Encapsulation of this prodrug into lipid nanoparticles (LNPs) enabled the successful intranasal delivery and brain targeting of the prodrug. This allows the central effects of LNP-encapsulated prodrugs without the unwanted peripheral effects; however, the efficacy of this approach on dopaminergic survival has yet to be evaluated.

Preclinical data have shown that low doses of mitochondrial uncoupling agent 2,4-dinitrophenol (DNP) can protect neurons and improve functional outcome in animal models of AD and PD, as well as epilepsy, and cerebral ischemic stroke by stimulating stress-response signaling pathways in neurons including those involving BDNF, the transcription factor cyclic AMP response element-binding protein (CREB), and autophagy [80]. A prodrug of DNP was synthesized to improve the pharmacokinetics profile of DNP, resulting in 20-fold lower C_max_ and 3-fold longer elimination time [81]. In a recent study DNP, and its prodrug MP201 were evaluated in the 6-hydroxydopamine PD animal model [82]. Animals receiving low doses of both the parent drug and the prodrug were protected against dopaminergic loss and motor dysfunction. Microglial reactivity and astrocytes reactivity that is caused by the 6-hydroxydopamine (6-OHDA) lesion were significantly reduced in mice that were treated with MP201 compared to mice in the placebo group. In addition, the pretreatment of mice with MP201 protected dopaminergic neurons against toxicity of mitochondrial complex I inhibition. Preclinical results of this prodrug approach, a modification of the mitochondrial uncoupling agent, might be the basis for the new disease-modifying therapies for PD [82].

Ultimately, PD patients often suffer from neurogenic orthostatic hypotension which leads to hypotension upon orthostatic challenge and depleted levels of norepinephrine. For this purpose, a prodrug of norepinephrine, droxidopa, was developed to increase concentrations of norepinephrin and dopamine in the body and brain, respectively. Increased levels of norepinephrine in the peripheral nervous system enable the body to maintain blood flow upon and while standing. Droxidopa can also cross the BBB and is converted to norepinephrine within the brain [83]. It was approved by the FDA in 2014 and is indicated for off-label therapy in neurogenic orthostatic hypotension in PD patients [8].

The development of new prodrugs that can pass through the BBB unaltered and exhibit translatable ADME (absorption, distribution, metabolism, elimination) profiles and pharmacological efficacies represent exciting challenges for medicinal chemists. We have presented several prodrug approaches that have been developed in recent years that resulted in products that demonstrated good pharmacokinetic profiles, sustained release of *L*-DOPA, and afforded reduced plasma level fluctuations. In the case of *L*-DOPA, prodrugs should have a fine balance of lipophilicity and hydrophilicity, be completely absorbed in the gut, be resistant to metabolic/chemical degradation, and deliver *L*-DOPA to the site of action intact. Importantly, ideal prodrug candidates should also provide stable plasma levels that result in continuous dopamine release, thereby avoiding side effects such as *L*-DOPA-related motor oscillations and dyskinesia.

## 3. The Use of Prodrugs for Infectious Diseases

Many microbial infections cause inflammation within the CNS through the activation of brain-resident immune cells and infiltration of peripheral immune cells. Such responses are necessary to protect the brain from lethal infections, but they also provoke neuropathological changes that lead to neurodegeneration. Indeed, studies suggest a potential involvement of enteroviruses and herpesviruses in the etiology of ALS [84], hepatitis virus in dementia [85], HIV in various neurological consequences [20], and most recently, severe acute respiratory syndrome coronavirus 2 (SARS-CoV-2) with the development of neurodegenerative pathology [86]. Prodrug approaches could provide better management of infectious diseases and potential prevention or intervention of the neuropathological consequences of infection.

The need for improvement of antiviral drug therapy grows rapidly with the recognition of more infectious viruses. Drug discovery methods have advanced significantly over the past 40 years, and the entire process of discovery can be broken down into sub-processes that include lead generation, lead optimization, and lead development. Lead generation includes numerous screening methodologies to the extent that hundreds to thousands of candidates can be screened against a particular target. These techniques present a new challenge, as many of the generated drug candidates have undesirable physicochemical features (poor permeability, poor solubility, and poor targeting) or likely to identify similar scaffolds. These drug candidates still require chemical modifications or use of different formulation techniques to accomplish an appropriate pharmacological effect and fulfill regulatory requirements. Another obstacle in antiviral drug development is antiviral drug resistance. To address these obstacles, prodrug design has proven to be one of the most effective approaches to more efficient treatments. On the other hand, in recent years, the enormous growth in the number of therapies that have been applied to infectious diseases has also necessitated a constant need for improvement to parental drugs. Thus, utilization of prodrug strategies to improve on first line drugs will afford products that exhibit lower dosing equivalents, longer dosing intervals, less toxicity, sustained release and/or retention times, and better efficacies.

### 3.1. Prodrug Therapies for Herpesviridae Infection: Herpes Simplex Virus (HSV) and Varicella Zoster Virus (VZV)

The development of acyclovir for the treatment of herpesvirus infections 50 years ago marked the age of antiviral therapy and provided one of the first examples of genuinely selective, efficacious antiviral drugs. It is still used today as an effective means for the treatment of herpes simplex virus (HSV) infections and varicella zoster virus (VZV), despite its poor bioavailability. The amino acid ester approach in designing nucleoside analogs has been implemented several times in the past to facilitate appropriate targeting of human intestinal peptide transporter 1 (PEPT1) to improve oral absorption. Indeed, a 5’valyl ester acyclovir prodrug, valacyclovir, was developed as a potent antiviral agent that increased oral bioavailability of acyclovir by three- to five-fold due to the permissive binding to and transport by PEPT1 [87]. Nevertheless, the exact mechanism of valacyclovir activation was unknown. The activation of amino acid ester prodrugs was considered non-specific, until the discovery of an enzyme that was responsible for its activation. Subsequently, valacyclovir conversion to its parent drug was shown to be dependent on the serine hydrolase, human valacyclovirase (hVACVase) enzyme [88]. This finding revealed one of the main steps in valacyclovir activation that is responsible for transformation to acyclovir and ultimate therapeutic effect. On this basis, other prodrugs were developed as well. For instance, the guanidino group on valacyclovir plays a significant physiological role due to its positive charge and strong electrostatic interactions with carboxylic groups (negatively charged). Numerous receptors show an attraction towards the amino acid, *L*-arginine, which contains a guanidino group that forms a strong bond due to the presence of carboxylate from the receptor active site. Prodrugs, designed to mimic the guanidino group, show specific targeting to that desired site [89]. However, drugs with guanidino groups typically exhibit low bioavailability following oral administration due to low passive diffusion that is attributable to an ionized drug in the intestinal environments. By masking the guanidino group, an amino acid ester prodrug [3-(hydroxymethyl)phenyl]guanidine (3-HPG) was created. This approach led to a double targeted prodrug design; the guanidino group for valacyclovirase-mediated activation and PEPT1 targeted intestinal absorption. This novel double prodrug approach demonstrated increased permeability compared to the parent drug and was attributable to PEPT1 transport of 3-HPG, and successful activation and conversion to the parent drug by valacyclovirase [89]. This unique design utilizes both activation and transport mechanisms to facilitate bioavailability and efficiency of the prodrug strategy. This example highlights the importance of identifying prodrug activation mechanisms to reduce the unnecessary testing during prodrug development and achieve higher predictability with greater performance and efficacy.

### 3.2. Prodrug Therapies for Human Immunodeficiency Virus (HIV)

Human immunodeficiency virus 1 (HIV-1) has been a significant challenge to global health for four decades since the first reported case [90]. The unique set of therapeutic challenges that HIV-1 poses include, but are not limited to rapid mutation, extensive incubation periods, and localization in tissue reservoirs [91]. However, even 40 years after the discovery of the virus, HIV-1 has evaded eradication from vaccines [92]. Furtherance of the therapeutic and prophylactic strategies comprised of daily oral antiretroviral drugs has been largely responsible for the significant decrease in the incidence of acquired immunodeficiency syndrome (AIDS) [93], but not eradication. This strategy is not without its own drawbacks and limitations such as low patient compliance, therapy exhaustion, poor availability of drugs, and cross-target toxicities that have led to the persistence of HIV infections [94]. A failure of the oral antiretroviral (ART) regimen could also be attributed to the poor tissue penetration of drugs, which leads to low drug levels in viral reservoirs such as lymph nodes, spleen, lung, and brain [95]. Persistent viral activity has been reported in reservoir sites even with direct parenteral administration of ART [95,96]. In the absence of promising avenues for vaccine development against HIV-1, long-acting antiretroviral prodrugs are promising candidates for chemovaccine development.

Recurrent issues with the clinical potential of several promising drugs against HIV-1 include inferior pharmacokinetics due to suboptimal physicochemical features; inferior absorption, distribution, metabolism, and excretion (ADME) characteristics; poor PK/PD profile; and formulation difficulty [9]. Some prodrug strategies involve the development of drug conjugates that are devoid of biological activity until the prodrug is activated by cleavage of the derivatizing promoiety through hydrolysis, which is mediated by either pH and/or metabolic enzymes to provide the pharmacologically-active parent drug.

The HIV viral life cycle involves several key steps which provide potential pharmacological targets for therapeutic intervention. The sequence of events in the viral life cycle includes surface binding of the virus to the cell surface of a CD4+ T cell or macrophage and subsequent fusion therein; capsid uncoating and reverse transcription of the viral RNA to cDNA followed by integration into the host genome; and eventually leading to transcription and translation of the viral DNA to produce new viruses that are released from the cell by budding [97]. Prodrug development for the treatment of HIV has also shown great promise. Most of the initiatives in ART prodrugs have been designed to inhibit proteins such as reverse transcriptase (the key protein in viral DNA synthesis), integrase (responsible for integration of the viral DNA into the host genome), and protease (responsible for cleavage of the protein and the generation of viral particles) [17]. Nucleoside reverse transcriptase inhibitors (NRTIs) were the first class of drugs to show utility against HIV-1 and remain one of the most prescribed ART therapeutics to date. This work has led to the development of eight drug candidates that have successfully obtained FDA approval for the treatment against HIV-1, with the most significant clinical outcomes reported with a prodrug of tenofovir that is known as tenofovir alafenamide (TAF) [98]. However, challenges that are associated with NRTIs such as short plasma half-life, low penetration into viral reservoir sites, and low membrane permeability due to its hydrophilic nature as well as the requirement to be activated in the metabolically-active triphosphate form limits the efficacy of this ART class [99,100]. The ineffective phosphorylation of nucleoside analogs due to their structural dissimilarities to natural nucleosides, downregulation of key kinases that are responsible for phosphorylation, and increased activity of cellular phosphatases have been attributed to decreased pharmacological activity of the NRTIs [98,101,102,103,104]. Moreover, the NRTIs offer a wide range of choices in prodrug development due to the availability of multiple sterically available functional groups for promoiety conjugation [105].

A key advancement was made in 1992 when McGuigan et al. developed novel aryl phosphate derivatives of azidothymidine using phosphorochloridate chemistry, thereby developing ProTide technology [106,107]. A ProTide (pronucleotide) is defined as a nucleoside aryl phosphate or phosphonate that is conjugated with an amino acid ester promoiety using a P-N bond linker [108]. Comprehensive reviews of the synthetic schemes for ProTides have been covered previously by Pradere et al. and Mehellou et al. [98,109]. Didanosine is a potent NRTI that was the second drug to be approved for treatment against HIV [110,111]. However, didanosine has been limited in application due to poor bioavailability of 20–40%, pH instability, and poor membrane permeability [111,112,113]. Yan et al. addressed these issues by the development of 5′-amino acid ester prodrugs of didanosine to utilize intestinal PEPT1-mediated transport and increase the acidic stability of didanosine [111]. The most significant success in the ProTide platform was achieved through the development of tenofovir alafenamide. Tenofovir is one of the most widely used drugs for HIV and hepatitis B treatment and is also approved for HIV pre-exposure prophylaxis (PrEP). Currently available are two forms of tenofovir; the older tenofovir disoproxil fumarate and the newer tenofovir alafenamide (TAF). While TDF showed promise in clinical outcomes, certain drawbacks were related to renal toxicities and reduced bone mineral density in TDF-treated patients [108,114]. TAF is the isopropylalaninyl monoamidate phenyl monoester prodrug of tenofovir and SP-diastereoisomer of TAF showed 1000-fold higher anti-HIV activity than TDF in vitro, thus suggesting the stereoselective nature of the phosphorylation cascade [114]. Phase 3 clinical trials comparing TAF and TDF found TAF to be non-inferior to TDF with lower incidences of adverse events and higher potency (GS-US-292-0112).

There has been a parallel initiative in the development of NRTI prodrugs beyond the ProTide platform. The principal focus has been to increase the hydrophobic-lipophilic character of the hydrophilic NRTIs to increase tissue penetration and apparent plasma half-life. Skanji et al. developed a glycerolipidic prodrug of didanosine that was incorporated into an orally administered liposomal delivery system, and showed significant delivery of the drug to viral reservoirs including testes, gut, and bone marrow [115]. Hillaireau et al. furthered these findings by developing squalenated didanosine and dideoxycytidine prodrug nanoassemblies that were stabilized with PEG and reported significant drug delivery to viral reservoirs [100]. Jin et al. developed a bolaamphiphilic prodrug by covalently conjugating zidovudine and didanosine by a deoxycholyl linker to direct this formulation to form self-assembled monolayer vesicles [116]. A novel development of a macrophage-targeted, macromolecular prodrug platform that was based on poly(l-lysine succinylated) emtricitabine was developed by Stevens et al. and found 7- to 19-fold higher concentrations in rat lymphatic tissues compared to controls that received parent emtricitabine intravenously [117]. Opting for a novel approach, Dalpiaz et al. developed a ursodeoxycholic acid-conjugated prodrug of zidovudine encased in a chitosan microparticle optimized for nasal administration [118]. Prodrug research in NRTIs has also led to repurposing of drugs that were previously thought to be pharmacologically inactive. D- and L-furano-D-apionucleosides were developed demonstrating antiretroviral activities by optimizing drug phosphorylation [119]. 

Protease inhibitors are drugs that inhibit viral aspartyl protease, a key enzyme that is required to cleave the HIV-1 Gag and Gag-Pol polyproteins that lead to the assembly and maturation of new viruses [120]. While protease inhibitors have shown promising results in terms of potency, their utility as first line therapeutic candidates has been limited due to poor bioavailability of oral formulations and multiple instances of adverse reactions including hyperlipidemia, insulin resistance, bone density loss, and cardiovascular disease [121]. A total of nine drugs of this class have been approved by the US FDA for clinical application for HIV-1 infection [122]. A comprehensive review pertaining to the utility of protease inhibitors as well as the scope for development of prodrugs [120]. Most prodrug development has utilized a secondary hydroxy group on the native drug to attach a promoiety [105]. Prodrug design strategies that were implemented for protease inhibitors include direct conjugation of promoiety to the native drug, conjugation of the promoiety to the drug by a cleavable linker, and O-N acyl migration where deprotonation of O-acyl precursors leads to an acyl group that is transferred from a hydroxyl moiety to the proximal amine [120]. However, many of these prodrug developments have reported a marked loss of antiretroviral activity due to the stable chemical modifications conjugated to a critical component of the pharmacophore [123]. Subbaiah et al. developed a (carbonyl)oxyalkyl linker-based amino acid prodrug of atazanavir and reported a five-fold higher AUC that was attained after oral administration in rats than that which was attained by the orally administered native drug [124]. However, PK assessment concluded at 24 h, thus the sustainability of antiretroviral activity over a longer time could not be substantiated. Qin et al. developed an alkyl ester prodrug library of lopinavir, loaded the prodrugs into artificial emulsions as well as chylomicrons, and delivered the nanoassemblies intravenously and orally to rats [125]. The group reported 7.2-fold higher drug levels in mesenteric lymph nodes as well as increased plasma half-lives of the various prodrugs. Despite these advancements with protease inhibitors, further research in this area is needed.

Pharmacological targets other than the key enzymes such as reverse transcriptase, integrase, and protease have been explored in depth. Fostemsavir (BMS-663068) is a phosphonooxymethyl prodrug of temsavir (BMS-626529), a novel small-molecule attachment inhibitor that targets HIV-1 spike protein gp120 [126,127]. Fostemsavir showed sustained antiretroviral activity up to 48 weeks in a clinical trial of 371 patients [128]. Protein kinase C (PKC) modulators show potent latency reversal activity and have shown promise with the development of tigliane and ingenane prodrugs of bryostatin 1 [129].

#### 3.2.1. Long-Acting Slow Effective Release Antiretroviral Therapy (LASER ART)

Many studies have previously described the delivery of ART therapeutics to viral reservoirs by using nanoparticles [130,131,132,133]; some attempted co-delivery of drugs in lipid nanoparticles [132]. However, the sustained antiretroviral efficacy of the drugs could only be achieved through combinations of the prodrug and nanoparticle technology platforms. Long-acting slow effective release antiretroviral therapy (LASER ART) is comprised of aqueous nanosuspensions of hydrophobic-lipophilic prodrug nanocrystals that are coated by stabilizing water-soluble surfactants [134]. LASER ART provides a two-step pathway entailing the sequential dissolution of the nanocrystals and the subsequent activation of the prodrug to provide the native drug. Intramuscular administration of LASER ART forms a primary depot at the injection site and secondary tissue depots from where the drug is slowly released to achieve sustained therapeutic active drug levels in systemic circulation and tissues [135]. The key parameters that allow for critical control of the pharmacokinetic and pharmacodynamic outcomes include, but are not limited to, the prodrug partition coefficient, pH stability of the nanoformulation, extent of perfusion at site of injection, and the potency of the native drug [135].

Our laboratory has successfully developed libraries of antiviral prodrugs to facilitate treatment and prevention of HIV and HBV infections. While NRTIs represent some of the most prescribed ART, their inherent physicochemical properties have limited their transformation into sustained release formulations. To overcome such limitations, our laboratory successfully produced ProTide libraries for darunavir [123], abacavir [136], emtricitabine [137,138], lamivudine [139], and tenofovir [140], all of which exhibit extended drug half-lives and efficient intracellular active metabolite delivery (Table 2). The most notable preclinical advancement was demonstrated for integrase inhibitors where a nanoformulated lipophilic ester prodrug of cabotegravir (NM2CAB) exhibited sustained plasma active drug levels above the protein-adjusted IC_90_ for a year in mice and rats (Figure 2) [135,141,142]. Previous studies with the first generation dolutegravir prodrug nanoformulation extended the half-life of the parent drug from hours to weeks [143]. The potential role of combinations of LASER ART and genome editing technologies that facilitate HIV eradication was evaluated in a chronic humanized mouse model for HIV-1 infection. While prodrug nanoformulations have shown potential for clinical translation, a total of 15 clinical trials have been conducted to assess the utility of prodrug formulations in treatment against HIV-1 and only fosamprenavir, tenofovir disoproxil fumarate (TDF), and tenofovir alafenamide fumarate (TAF) have been approved as daily oral therapies (Table 2) [17]. This further highlights the need for the development of novel prodrugs that facilitate less frequent dosing intervals to manage chronic conditions.

#### 3.2.2. Prodrug Therapies for Human Immunodeficiency Virus (HIV)-Associated Neurocognitive Disorders (HAND)

Life expectancy of HIV-1-infected people has increased significantly due to ART. Nevertheless, end-organ disease persists with almost constant low level infection eliciting, for instance, HIV-associated neurocognitive disorders (HAND); indeed, up to 50% of ART-treated HIV-1-infected people develop HAND [142]. The HAND spectrum includes asymptomatic neurocognitive impairment, mild neurocognitive disorder, and HIV-associated dementia. The long-term health prognosis of aging with controlled HIV infection and HIV-associated neurocognitive disorder (HAND) remains unclear.

Zidovudine (AZT) is a nucleoside reverse transcriptase inhibitor (NRTI) that is used along with other medications to treat HIV infection. Most commonly, zidovudine is given to HIV-positive pregnant women to reduce the chance of passing the infection to the baby. It is a substrate of active efflux transporters (AETs) that extrude the drug from the CNS and macrophages. The prodrug approach in which AZT and ursodeoxycholic acid (UDCA) were conjugated to create a new prodrug molecule that evades the AET system, demonstrated the potential of UDCA to behave similar to an AZT carrier to the CNS and into macrophages. The resulting molecule, UDCA-AZT is indeed, very permeable, and remains in murine macrophages with an efficacy that is 20-fold higher than parent AZT. This approach could be potentiated through formulations to allow intranasal administration aiming to provide faster/easier brain uptake. Intranasal instillation of chitosan chloride-based microparticles containing UDCA-AZT were able to increase the dissolution rate of UDCA-AZT, reduce water uptake with respect to its original physical mixture, and produce better prodrug uptake into the cerebrospinal fluid of rats where the prodrug can then act as an AZT carrier into the macrophages [118,144].

### 3.3. Prodrugs Therapies for Hepatitis B and C Infection

Globally, hepatitis B Virus (HBV) and C (HCV) are viral pathogens that are responsible for 296 million and 58 million cases, respectively. It is commonly experienced as an illness of the liver, which, in some circumstances, can lead to a chronic infection with the potential to induce progressive fibrosis, cirrhosis, or even cancer of the liver [145]. There is also an association between hepatitis B and C virus (HBV and HCV) infections and dementia.

Several therapeutics that are currently in use can slow hepatocellular infection and/or insult. Derivatives of tenofovir are commonly used for this indication. As mentioned in the previous section, two tenofovir prodrugs, TDF and TAF, are currently marketed. TAF was designed to improve plasma stability of the parent drug and to enable more efficient delivery to hepatocytes compared to its predecessor TDF [146]. Indeed, TAF proved to be as potent as TDF at much lower doses due to selective uptake of tenofovir in hepatocytes [146,147]. The benefit of this prodrug is driven by decreased dosages that allow mitigation of non-specific toxicity, making TAF safer for use in patients than TDF for long-term treatment [148]. This was observed through the differential effect on the mean degradation of bone mineral density between patients who were treated with TAF as opposed to TDF (0.33% to 2.551%, respectively) as well as improvement in renal impairment during Phase 3 clinical trials [147].

Another prodrug that is currently undergoing clinical trials for hepatitis infection is pradefovir (adefovir dipivoxil), presented in the Table 2 [149]. Pradefovir, is a cyclodiester antiviral prodrug that has activity against chronic hepatitis B infection by targeting the HBV DNA polymerase [150]. It is specifically metabolized in the liver by hepatic enzymes, mainly CYP 3A4, to its parent drug, adefovir, which is then phosphorylated by cellular kinases to its activated form, adevofir diphosphate. By competing with the natural substrate deoxyadenosine triphosphate (dATP), the diphosphate form is incorporated into viral DNA and blocks the RNA-dependent DNA polymerase, causing DNA chain termination and consequent inhibition of HBV replication. Much like TAF, pradefovir was found to be safer than TDF, with similar levels of HBV DNA reduction at much lower doses (30 mg, 45 mg, 60 mg, and 75 mg in comparison to 300mg of TDF) [151]. While not yet approved by the FDA, it is currently undergoing Phase 3 clinical trials and shows potential for a novel treatment option for HBV infection [152].

Hepatitis C virus (HCV), much like HBV, is a virus that targets the liver, but unlike HBV, patients who contract this virus have a higher risk of developing cirrhosis and hepatocellular carcinoma. Although highly effective disease-acting antiviral compounds (DAAC’s) are available, they are effective for only a subset of patients [153]. This is due to the genetic heterogeneity of HCV that presents as several different forms, some of which produce more serious outcomes than other forms, and in turn, makes drug development even more challenging [154]. AT-527 is a prodrug with a unique mechanism of action and dual target effects. Mainly, AT-527 targets RNA-dependent RNA polymerase (RdRp) chain termination and Nidovirus RdRp associated nucleotidyl transferase (NiRAN) inhibition, which has the potential to create a high barrier to resistance. This allows for the potential to create a high barrier to resistance, and has been shown to be very efficacious in clinical trials for HCV (Table 2) [155]. The benefits of this prodrug compared to other therapies that are available are prolonged activity, potential avenue for treatment among difficult to treat subsets of patients, and selective delivery that is facilitated through the metabolized intermediates of prodrugs.

### 3.4. Prodrug Therapies for COVID-19

In less than two years, severe acute respiratory syndrome coronavirus 2 (SARS-CoV-2), the etiological agent of coronavirus disease 2019 (COVID-19), has infected over 350 million people, killed over 5.5 million people, overwhelmed the global healthcare infrastructure, and unprecedentedly paused the world’s workforce and economies. Therefore, it is imperative to prepare downstream medical opportunities for therapeutic interventions [160]. While there are currently highly efficacious avenues for preventative care (upstream medicine), drugs that increase health outcomes in patients that are suffering from severe forms of COVID-19 (downstream medicine) remain undefined [161]. However, it should be mentioned that, most recently in December 2021, the FDA issued an emergency use authorization (EUA) for Paxlovid^®^ (nirmatrelvir and ritonavir tablets, co-packaged) for oral use for the treatment of mild–moderate coronavirus disease (COVID-19) in adults and pediatric patients with positive SARS-CoV-2 testing, and who are at high risk for progression to severe COVID-19.

One of the promising prodrugs for COVID-19 treatment is molnupiravir, a prodrug of N4-hydroxycytidine (Table 2). Its mechanism of action promotes widespread mutations in the replication of viral RNA by RNA-directed RNA polymerase. The prodrug is metabolized into a ribonucleoside analog, β-D-N4-hydroxycytidine 5′-triphosphate, which resembles cytidine and is incorporated into newly synthesized RNA in place of cytidine during replication. In the clinic, molnupiravir was shown to decrease the risk of hospital admission or lethal outcome by around 30% in non-hospitalized patients with mild to moderate COVID-19 infection, and a high risk of poor outcome [158]. Most recently, molnupiravir was approved by UK Medicines and Healthcare products Regulatory Agency (MHRA) [162].

Remdesivir, is a ProTide which inhibits viral RNA synthesis (Table 2), and was developed as a medicine for treating other RNA-based viruses, including Ebola virus (EBOV) and other viruses within the *Coronaviridae* family [163]. A double-blind, randomized, placebo-controlled trial of intravenous remdesivir in adults that were hospitalized with lower respiratory tract COVID-19 infection showed superiority to placebo in shortening recovery periods in people that were hospitalized with COVID-19. Remdesivir was recently approved by the FDA for use in SARS-CoV-2 infections [157].

Another promising candidate for the treatment of COVID-19 is AT-527. As described in the previous section, AT-527 is an orally administered double prodrug of a guanosine nucleotide analog, that was previously proven to be highly effective against HCV. It was found to be a highly potent drug against SARS-CoV-2 infection in vitro, yielding 90% inhibition of viral replication in human epithelial airway cells at a concentration of 0.47 µM [156]. Notably, in comparison to remdesivir, AT-527 can be administered orally and is currently undergoing Phase 3 clinical trials.

Altogether, the prodrug approach has been shown to be very successful in modifying nucleoside analogues for numerous infectious diseases, whether it is overcoming resistant barriers, passing BBB, improving bioavailability following oral administration, or reducing the side effects that are associated with the parent drug.

## 4. Discussion

To impact neurodegenerative and neuro-infectious disease outcomes, medicines must penetrate the BBB and enter the CNS. However, the BBB has considerable structural and functional complexities and represents a significant obstacle for brain-directed drug delivery. This is especially noteworthy at the early stages of disease when the BBB is intact. Over the years, several approaches were used to improve BBB permeability. This was done through several approaches including carrier-mediated transport in combination with endogenous ligands, the use of lipophilic carriers to improve physicochemical drug properties, the combination of two distinct pharmacophores that were chemically-linked together to improve the drug delivery properties, and various drug delivery and formulation approaches [65]. Additionally, successful brain targeting can also be achieved through a combination of prodrugs and drug delivery systems. Indeed, chemical and enzymatic protection that is operated by the carrier, joined with the ability of the prodrug to cross the BBB, has allowed slow and sustained release, thus reducing the plasma fluctuation for better control of the disease [164]. That, alongside suitable formulation that allows a particular route of administration, such as inhalation to provide a promising avenue for the treatment of neurodegenerative and neuro-infectious diseases.

Another obstacle that modern therapeutics face for treating neurodegenerative diseases is that these disorders are often multifactorial. Various pathogenic pathways that encompass protein misfolding, aggregation, mitochondrial dysfunction, oxidative stress, free radical formation, and phosphorylation impairment as well as infection, contribute to disease progression. The need for novel drug entities that can target multiple processes that are involved in disease development and progression is warranted. The improvement of existing therapeutics strategies, or a combination of several drug entities is crucial to address several causative aspects of such disorders. The prodrug approach is one of the most promising avenues for enhancing pharmaceutical, pharmacokinetic, and pharmacodynamic properties of hydrophilic compounds, such as anti-Parkinson drugs (DA and LD) [67]. One beneficial approach in such conditions is also the co-drug approach, where a single chemical moiety is comprised of two separate pharmacophores. In this approach, a molecule having an antioxidant effect could be coupled with a drug that restores dopamine levels in the brain, thus permitting compensation of the dopaminergic effect while simultaneously preventing neuronal depletion by targeting oxidative stress [165].

With the treatment of infectious diseases, great advances have been made in the prodrug design. Long-acting formulations of prodrugs for HIV and hepatitis will set the stage for the development of drugs for other infectious diseases that lack suitable therapeutic interventions such as COVID-19. We and others have made significant advances in extending the apparent half-life of numerous ART as well as improving their safety of administration and ease of manufacturing. Indeed, the innovative prodrug molecular design and improved physicochemical properties have been shown to affect pharmacokinetic and targeting properties of native drugs and have repeatedly been used to treat a spectrum of infectious and/or degenerative disorders [139,140,141,142,147,151,156]. Thus, continuous research attempts and in-depth knowledge of molecular and cellular mechanisms underlying pathogen-mediated neuronal damage may establish the way to find new preventative and therapeutic strategies that are aimed at reducing the advancement of these devastating pathologies.

## 5. Conclusions

The prodrug strategy is designed to improve parent drug properties and enhance drug transport across physiological barriers and/or allow encapsulation for enhanced delivery systems/formulations. Microbial-induced neuroinflammation, in many cases, may lead to neurodegenerative pathology. This work provides an overview of the prodrug approach as a tool for treating infections with consequent prevention of developing neurodegenerative disease pathology, and for preclinical and clinical development of prodrugs that are associated with the treatment of common neurodegenerative diseases. We deliver examples of possible uses and advantages of existing prodrugs, such as improving BBB permeability, overcoming drug resistance, enhancing drug targeting abilities, decreasing rapid metabolism, increasing treatment safety, and enabling encapsulation into formulations and ease of manufacturing. Some drawbacks of the prodrug approach include limitations that are connected with variant formulation abilities, inconsistent rate/extent of prodrug hydrolysis, and low selectivity. Nevertheless, the prodrug approach is continuously being developed and improved, and is a crucial piece in the drug discovery toolbox, which is likely to grow further in the future.

## Figures and Tables

**Figure 1 pharmaceutics-14-00518-f001:**
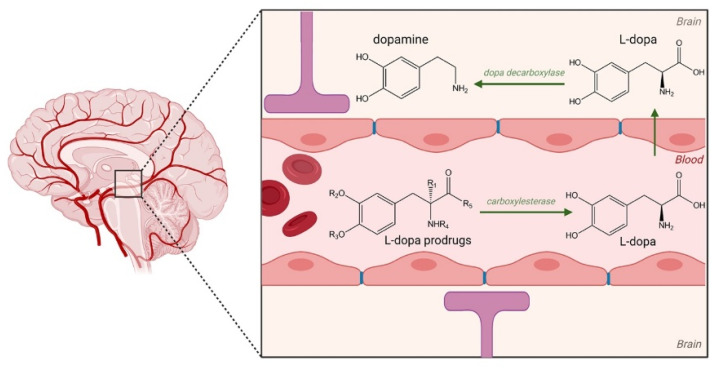
Bioconversion of levodopa (*L*-DOPA) prodrug to *L*-DOPA by enzyme carboxylesterases in the blood (case of XP21279). *L*-DOPA then passes through the blood-brain barrier (BBB) and permeates into the brain where it is converted to dopamine via DOPA decarboxylase. Created using BioRender.com (Accessed on 2 January 2022).

**Figure 2 pharmaceutics-14-00518-f002:**
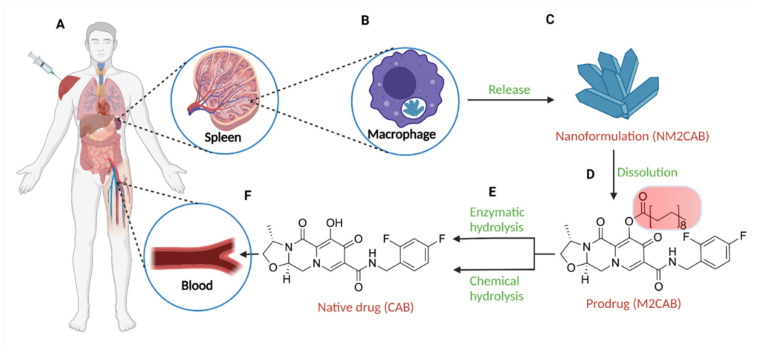
Mechanism for in vivo activity of LASER ART. (**A**) Intramuscular injection of cabotegravir prodrug nanoformulation (NM2CAB). (**B**) Lymphoid organs such as spleen act as secondary depots for the nanoformulation with internalization of NM2CAB by splenic macrophages. (**C**) The nanoformulation is slowly released from the macrophages and enters the extracellular matrix. (**D**) Dissolution of the nanocrystals provides free prodrug M2CAB. (**E**) Enzymatic hydrolysis and alkaline pH release the promoiety from the native drug. (**F**) The pharmacologically active native drug (CAB) enters the systemic circulation and elicits its activity as a potent integrase inhibitor. Created using BioRender.com (Accessed on 2 January 2022).

**Table 1 pharmaceutics-14-00518-t001:** Structures of the selected prodrugs that were developed for AD therapy.

**R13: 7,8-DHF prodrug** [54]	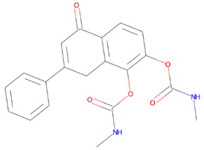
**Memit** [46]	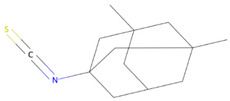
**ALZ801** [63]	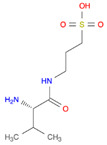

**Table 2 pharmaceutics-14-00518-t002:** Prodrug molecules used for HIV, Hepatitis B, and COVID-19.

HIV	Hepatitis B	COVID-19
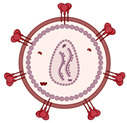	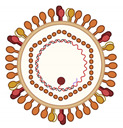	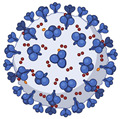
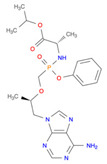 Tenofovir alafenamide (TAF) [146,147]	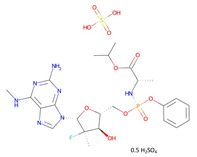 AT-527 (Bemnifosbuvir) [156]	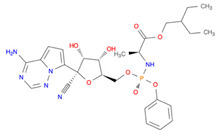 Remdesivir [157]
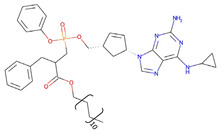 Prodrug of abacavir (M3ABC) [136]		
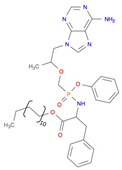 Prodrug of tenofovir (M1TFV) [140]		
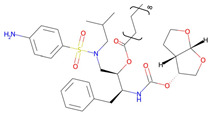 Prodrug of darunavir (M2DRV) [123]	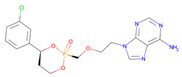 Pradefovir [149,150]	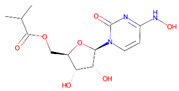 Molnupiravir [158]
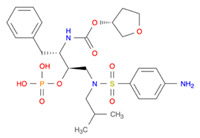 Fosamprenavir [159]		

## Data Availability

Not applicable.

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
