# Peer review of "Prodrug Therapies for Infectious and Neurodegenerative Diseases"

_pharmaceutics, 2022, doi:10.3390/pharmaceutics14030518_

Round 1

Reviewer 1 Report

A few comments that may add to the quality of the Manuscript:

  1. first sentence in the Abstract should be written later because now it somewhat repeats itself.
  2. They, as a group - please specify instead of using a pronoun
  3. This sentence doesn't make sense  This comes as a surprise for infectious and degenerative diseases, in particular, as they both represent top leading causes of morbidity and
    mortality world-wide - just because a disease is common, doesn't mean new prodrugs should be developed, drugs yes but they do not need to be prodrugs
  4. More deaths does not mean greater prevalence, please clarify the following Women are disproportionately affected as they now represent 65% of AD deaths
  5. Rather than explaining the prevalence of diseases, explain the pitfalls of current treatments or lack of them in the introduction. Great prevalence and no treatment requires new drugs, not necessarily new prodrugs
  6. The introductory parts should be shortened and more figures should be added with a medicinal chemistry approach, explaining what was achieved with development of prodrug i.e. shielding a specific functional group

I believe that this paper need a different approach, more medicinal chemistry and less of pathophysiology, as it has not been clearly explained for most neurodegenerative diseases.

Author Response

Thank you for your favorable review!

Point 1: First sentence in the Abstract should be written later because now it somewhat repeats itself.

 Response. Thank you. As per Reviewer’s suggestion, the first sentence of the abstract is now amended (Page 1, Lines 16-17).

Point 2: They, as a group - please specify instead of using a pronoun

Response. Thank you! This error was corrected (Page 1, Lines 36-38).

Point 3: This sentence doesn't make sense.  This comes as a surprise for infectious and degenerative diseases, in particular, as they both represent top leading causes of morbidity and
mortality world-wide - just because a disease is common, doesn't mean new prodrugs should be developed, drugs yes but they do not need to be prodrugs

Response. As per reviewer’s comments, we amended this component of the Introduction. We highlighted the needs for therapeutic development and appreciate that such improvements may not be prodrugs (Page 2, Lines 81-97).

Point 4: More deaths does not mean greater prevalence, please clarify the following women are disproportionately affected as they now represent 65% of AD deaths

Response: We agree that the greater prevalence does not mean more deaths. To clarify this, we rewrote the section (Page 2, Line 69-70).

Point 5: Rather than explaining the prevalence of diseases, explain the pitfalls of current treatments or lack of them in the introduction. Great prevalence and no treatment requires new drugs, not necessarily new prodrugs.

Response. We agree with the reviewer. Disease prevalence is not the driver for developing new prodrugs. We have addressed this need by adding a section in the introduction describing the pitfalls of the current treatments, and the need for novel medicines for addressing highly prevalent diseases. This can be achieved by facilitating discovery and development of new therapies (Page 2, Lines 80-91).

Point 6: The introductory parts should be shortened and more figures should be added with a medicinal chemistry approach, explaining what was achieved with development of prodrug i.e. shielding a specific functional group

Response. Thank you. Accordingly, we have incorporated numerous changes that reflect medicinal chemistry and functional group shielding approaches for prodrug development (Page 4, Lines 155-161; 166-168; 192-196; Page 5, Line 207-209; 221-224; Page 6, Line 259-262, Page 7, Lines 289-391). Thanks.

Point 7: I believe that this paper need a different approach, more medicinal chemistry and less of pathophysiology, as it has not been clearly explained for most neurodegenerative diseases.

Response. We thank the reviewer for his/her insightful comments. Amendments were made in the text that better reflect medicinal chemistry. This is, in particular for the neurodegenerative disease sections which have been amended in yellow (Page 4, Lines 155-161; 166-168; 192-196; Page 5, Line 207-209; 221-224; Page 6, Line 259-262, Page 7, Lines 289-391).

Reviewer 2 Report

In this Review manuscript, the authors summarized the recent developments and applications of prodrugs for treating neurodegenerative, inflammatory, and infectious diseases. The characteristics and existing problems of prodrugs associated with each disease were described. This review provided a meaningful guidance for prodrugs designing. Thus, I recommended the publication of the manuscript in Pharmaceutics after minor revision. List of comments are shown below:

1.In the abstract, the authors claimed that “Their interplay is reflected in the fact that many microbial infections are mitigating factors for neurodegenerative diseases independent of aging.” Is there a link between infectious diseases and neurodegenerative diseases? If so, the relationship between infectious and neurodegenerative diseases should be further discussed in main text of the article.

2.On page 4, the “cysteine-dependent mechanism” is suggested to be added a reference article or explained briefly.

3.The structure of prodrug derivative R13 should be added in table 1.

Author Response

Thank you for your favorable review!

Overview: In this Review manuscript, the authors summarized the recent developments and applications of prodrugs for treating neurodegenerative, inflammatory, and infectious diseases. The characteristics and existing problems of prodrugs associated with each disease were described. This review provided a meaningful guidance for prodrugs designing. Thus, I recommended the publication of the manuscript in Pharmaceutics after minor revision. List of comments are shown below.

Response: Thank you!

Point 1: In the abstract, the authors claimed that “Their interplay is reflected in the fact that many microbial infections are mitigating factors for neurodegenerative diseases independent of aging.” Is there a link between infectious diseases and neurodegenerative diseases? If so, the relationship between infectious and neurodegenerative diseases should be further discussed in main text of the article.

Response. Thank you for your insightful critique! We have further described and elucidated the interplay between neurodegenerative and infectious disease throughout the manuscript (Page 1, Line 25-27; Page 2, Lines 63-66; 80-91; Page 8, Lines 338-346; Page 13, Lines 577-579, Page 17, Lines 732-736).

Point 2: On page 4, the “cysteine-dependent mechanism” is suggested to be added a reference article or explained briefly.

Response. We have provided detailed explanation of the “cysteine-dependent mechanism” in the activation of the Memit prodrug (Page 4, Lines 155-160). Thanks.

Point 3: The structure of prodrug derivative R13 should be added in Table 1.

Response. The structure of the prodrug derivative of R13 is highlighted in yellow in the Table 1 (Page 5). Thank you.
